# Characterization, High-Density Fermentation, and the Production of a Directed Vat Set Starter of *Lactobacilli* Used in the Food Industry: A Review

**DOI:** 10.3390/foods11193063

**Published:** 2022-10-02

**Authors:** Yun Lu, Shuqi Xing, Laping He, Cuiqin Li, Xiao Wang, Xuefeng Zeng, Yifeng Dai

**Affiliations:** 1Key Laboratory of Agricultural and Animal Products Storage & Processing of Guizhou Province, Guizhou University, Guiyang 550025, China; 2Department of Brewing Engineering, Moutai University, Renhuai 564507, China; 3College of Liquor and Food Engineering, Guizhou University, Guiyang 550025, China; 4School of Chemistry and Chemical Engineering, Guizhou University, Guiyang 550025, China

**Keywords:** *Lactobacilli* strains, probiotics, characterization, performance improvement, production of DVS starter

## Abstract

*Lactobacilli* have been widely concerned for decades. Bacteria of the genus *Lactobacillus* have been commonly employed in fermented food to improve the appearance, smell, and taste of food or prolong its shelf-life. They comprise 261 species (by March 2020) that are highly diverse at the phenotypic, ecological, and genotypic levels. Some *Lactobacilli* strains have been documented to be essential probiotics, which are defined as a group of living microorganisms that are beneficial to the health of the host when ingested in sufficiency. However, the characterization, high-density fermentation, and the production of a directed vat set (DVS) starter of *Lactobacilli* strains used in the food industry have not been systematically reported. This paper mainly focuses on reviewing *Lactobacilli* as functional starter cultures in the food industry, including different molecular techniques for identification at the species and strain levels, methods for evaluating *Lactobacilli* properties, enhancing their performance and improving the cell density of *Lactobacilli*, and the production techniques of DVS starter of *Lactobacilli* strains. Moreover, this review further discussed the existing problems and future development prospects of *Lactobacilli* in the food industry. The viability and stability of *Lactobacilli* in the food industry and gastrointestinal environment are critical challenges at the industrial scale. The new production equipment and technology of DVS starter of *Lactobacilli* strains will have the potential for large-scale application, for example, developing low-temperature spray drying, freezing granulation drying, and spray freeze-drying.

## 1. Introduction

*Lactobacilli* are Gram-positive rod, non-spore-forming, catalase-negative bacteria that commonly colonize the human intestine and have essential physiological functions in the human body [1,2]. Microscopically, these bacteria represent non-motile, thin rods that differ from long to short. Sometimes, they are present in coryneform, bent morphology, or chains. *Lactobacilli* comprise 261 species (by March 2020) that are highly diverse at the phenotypic, ecological, and genotypic levels [3]. Moreover, *Lactobacilli* have been documented to be important probiotics, defined as a group of living microorganisms that are beneficial to the host’s health when ingested in sufficiency [4]. 

*Lactobacilli* may be added as starters, which are used to ferment and produce specific changes in the chemical composition and sensory properties of foods [5]. It can produce amylase, protease, dehydrogenase, decarboxylase, β-glucosidase, and peptidase during the fermentation, thereby can be widely used in the food industry for the production of yogurt [6], cheese [7], sourdough [8], sausages [9], cucumber pickles [10], olives [11], sauerkraut [12], and so on. However, there is some diversity in the *Lactobacilli* used in fermented food, depending on the food matrix. One example is *Lactobacillus plantarum*, which is used as a starter culture in meat and wine (malolactic) fermentation [5], whereas *L*. *bulgaricus* can be found as the primary starter culture in yogurt fermentation [13]. 

The nutritional quality may increase during fermentation by *Lactobacilli* strains because the hydrolytic enzymes produced by the bacteria hydrolyze the complex macromolecule into simpler forms [14]. The characteristics of *Lactobacilli* strains in fermented food mainly include antimicrobial activity [15], acid and bile tolerance [16], gastrointestinal transit tolerance [17], cell surface properties (such as auto-aggregation, co-aggregation, and bacterial hydrophobicity) [18], metabolic products (such as bacteriocin, organic acids, fatty acids, hydrogen peroxide, bioactive peptides, and cell wall components) [19], and the evaluation of safety such as antibiotic resistance [20]. Additionally, *Lactobacilli* strains with high activity and stability can meet the rapid development of industrial food production [16]. The most common methods for long-term storage of *Lactobacilli* strains are freezing, spraying, and fluidized bed drying [21,22]. However, the latest progress, existing problems, and prospects on characterization, high-density fermentation, and directed vat set starter production of *Lactobacilli* used in the food industry still lacks systematic excavation. This paper mainly focuses on reviewing *Lactobacilli* as functional starter cultures in the food industry, including different molecular techniques for identification at the species and strain levels, methods for evaluating *Lactobacilli* properties, improving their performance and improving the cell density of *Lactobacilli*, the production techniques of DVS starters of *Lactobacilli* strains and future perspectives to be overcome in this area. Moreover, this review further discusses the existing problems and future development prospects of *Lactobacilli* in the food industry. 

## 2. Characterization of *Lactobacilli* Strains

### 2.1. Screening out Lactobacilli Strains

*Lactobacilli* used in the fermented food industry are diverse and many (Table 1). Due to their excellent fermentation performance, they are extensively used to ferment food based on various raw materials, including milk, meat, cereals, fruits, vegetables, and seafood. Their commercial products, including probiotics, have ample market space [23]. So, screening out one or several new strains with excellent fermentation performance and potential probiotic properties is very meaningful work. High-throughput screening technology is a method for the quick selection of certain strains of *Lactobacilli* species with outstanding performance (such as extracellular polysaccharides, bacteriocin, gamma amino acid, butyric acid, short-chain fatty acid, etc.) [24]. These *Lactobacilli* strains can be traditionally isolated from a wide range of sources, such as human and animal mucosal membranes, plants or material of plant origin, and fermented food.

We constructed a phylogenetic tree containing 64 *Lactobacilli* species based on the related *Lactobacilli* 16S rRNA gene sequence derived from National Center for Biotechnology Information (https://www.ncbi.nlm.nih.gov/) (Figure 1). It shows that the *Lactobacilli* species used in the food industry are not clustered in a particular branch but are almost evenly distributed in all branches of the *Lactobacilli* phylogenetic tree. Based on this, the other species in this genus may also have potential use value in the field of food, livestock breeding, and medicine. Given that *Lactobacilli* have strain specificity, the traditionally safe *Lactobacilli* strains cannot ensure the safety of all strains in the species. Thus, the identification and detection of *Lactobacilli* are of significance. 

Generally, microorganisms always contain desirable and undesirable characteristics. A qualified candidate for use in the food industry initially requires a desirable characterization, especially safety. For *Lactobacilli* strains, we expect to select strains with probiotic properties. These characteristics of new *Lactobacilli* strains need to be evaluated by in vitro and in vivo experiments to understand their potential probiotic properties and industrial uses. Table 2 summarizes in vitro evaluation assays of *Lactobacilli* strains used in screening.

### 2.2. Identification and Safety Assessment of Lactobacilli Strains

The safety status of *Lactobacilli* used in food production has become the focus of the attention of manufacturers and government departments. Although European Qualified Presumption of Safety regulations have identified microorganisms that can be safely used in food [91], some safety aspects must be evaluated before the newly screened cultures are applied to fermented food. For a newly screened *Lactobacilli* strain, its inherent characteristics need to be tested in vitro. First, we need to identify the isolate to make sure it is what we want and safe, and it can preliminarily predict whether the strains of this genus can be fully put into production [44]. Figure 2 describes some essential characterization contents of the new *Lactobacilli* strain. 

The identification methods of *Lactobacilli* use phenotypic methods and molecular identification methods. In contrast to phenotypic approaches, molecular identification and characterization tools can distinguish even between closely related groups of species, which are indistinguishable based on phenotype, which is far more consistent, quick, trustworthy, and reproducible [46,47]. The most commonly employed molecular techniques for the identification of *Lactobacilli* can be divided into two groups: species-specific identification techniques (including amplified ribosomal DNA restriction analysis (ARDRA) and 16S and 23S rRNA sequencing) and strain-specific identification techniques (including ribotyping, restriction enzyme analysis (REA) with pulsed-field gel electrophoresis (PFGE), genetic probes/DNA dot blot, multiplex PCR using specific primers, randomly amplified polymorphic DNA (RAPD), amplified fragment length polymorphism (AFLP), PCR-denaturing gradient gel electrophoresis (DGGE), and fluorescent in situ hybridization (FISH)) [92,93,94,95,96,97,98,99,100,101,102,103,104,105,106]. As an overview, these methods and evaluations are listed in Table 3. Taxonomy and phylogeny of the genus *Lactobacillus* have been recognized as rather complicated, because of a great number of species with a diverse group of species [3]. From the Table 3, it is clear that for reliable species determination within this genus, a polyphasic approach based primarily on one or more molecular methods is required. Additionally, the International Committee on Systematic Bacteriology has acknowledged polyphasic taxonomy as a trustworthy method for describing species and revising the current nomenclature of specific bacterial groupings. In a recent study, Zheng et al. [3] proposed reclassification of the genus *Lactobacillus* into 25 genera including the emended genus *Lactobacillus*, which includes host-adapted organisms that have been referred to as the *Lactobacillus* delbrueckii group, Para*lactobacillus* and 23 novel genera including *Holzapfelia*, *Amylolactobacillus*, *Bombilactobacillus*, *Companilactobacillus*, *Lapidilactobacillus*, *Agrilactobacillus*, *Schleiferilactobacillus*, *Loigolactobacilus*, *Lacticaseibacillus*, *Latilactobacillus*, *Dellaglioa*, *Liquorilactobacillus*, *Ligilactobacillus*, *Lactiplantibacillus*, *Furfurilactobacillus*, *Paucilactobacillus*, *Limosilactobacillus*, *Fructilactobacillus*, *Acetilactobacillus*, *Apilactobacillus*, *Levilactobacillus*, *Secundilactobacillus*, and *Lentilactobacillus*. This reclassification reflects the phylogenetic position of the micro-organisms and groups *Lactobacilli* into robust clades with shared ecological and metabolic properties that can anticipate the addition of new species shortly.

Then, the relevant antibiotic susceptibility is usually determined and evaluated according to the protocol provided by the European Food Safety Agency (EFSA) [107]. Microdilution broth tests on test tubes, disk diffusion [108], and commercial ready-to-use kits [109] have been used to determine the physical sensitivity of known antibiotics to newly screened strains. Hemolytic activity was also investigated [49]. The production of various enzymes should also be evaluated. Maybe they are the cause of pathogenicity. Strains should be tested for known human toxins (e.g., cytolysin) by appropriate in vitro analysis. Detecting the toxicity of pathogenic genes and metabolites is also conducted; several *Lactobacilli* can decarboxylate and reduce amino acids in food to produce biogenic amines, which can cause poisoning symptoms if they accumulate in excess amounts in the body [110]. These in vitro experimental analyses are simple and rapid in determining the safety of a newly screened strain and avoid the use of harmful strains. For example, a hemolytic and toxin-producing strain can easily be excluded from further analysis [49]. The false negative strains created by in vitro experimental research are concerning. Therefore, further in vivo experiments are needed, including animal models and clinical applications [11,82]. 

### 2.3. Potential Probiotic Functionalities of Lactobacilli Strains

Some *Lactobacilli* have been reported as strains with high probiotic potential and support efforts to improve probiotic quality, such as *L*. *salivarius* strains BCRC14759 and BCRC 12574, with the highest exopolysaccharide production [111], *L*. *johnsonii* ZLJ010, with better adaptation to the gut environment and its probiotic functionalities [112], and *L*. *helveticus* D75 and D76 that can inhibit the growth of pathogens and pathobionts [20]. However, *Lactobacilli* strains in the probiotic market are still limited, and *Lactobacilli* strains with potential probiotic properties should be explored. An important aspect is to evaluate the selected *Lactobacilli* in vitro and find their probiotic potential. Some in vitro probiotic performance evaluations of the strains include survival under stress (low pH, high bile salt, high osmotic pressure, high oxygen, oxidation, starvation, etc.), adhesion ability, and antibacterial, antioxidation, cholesterol-lowering, and anticancer activities (Table 2).

As probiotics, *Lactobacilli* colonizing the intestine to reach 1 × 10^6^ CFU is necessary for its probiotic effect [113]. *Lactobacilli* can survive in the robust acid environment in the gastric juice and high bile salt concentration in the small intestine, which are two criteria for screening good probiotic *Lactobacilli* strains. The acid and bile salt tolerance of *Lactobacilli* strains use the rate of viable bacteria incubated in various acid pH environments as an indicator in in vitro assays. Additionally, many studies conducted artificially simulated gastric juice tolerance and animal model tests of probiotic *Lactobacilli* [23,72,76]. The survival rate was used as an index to evaluate probiotic *Lactobacilli*’s acid and bile salt tolerance. 

Adhesion is another of the essential characteristics of probiotic bacteria that contributes to the colonization of probiotics in the gastrointestinal tract [67]. The ability of the bacteria to stick with hydrocarbons determines the extent of adhesion to the epithelial cells in the gastrointestinal tract, known as cell surface hydrophobicity [63]. The direct method of cell surface hydrophobicity of bacteria is to determine the change of absorbance of the supernatant of bacterial cell solution at 600 nm after treatment with hydrocarbons such as n-hexadecane and toluene. More precisely, the adhesion of *Lactobacilli* strains to mucin has also been determined [65]. Moreover, commercial kits for determining these mucins have been reported and can be used for high-throughput screening [66].

Intestinal epithelial cell (IEC) lines are often presumed to better represent conditions in the tissues of the GIT. Several studies have been conducted using human epithelial cell lines (such as HT-29, HT-29MTX, and Caco-2) to screen the adhesion of probiotic strains [67]. Additionally, other studies have focused on the self-aggregation of probiotics [18], which is also related to adhesion.

*Lactobacilli* strains can secrete lactic acid and other organic acids, lowering the environment’s pH and thereby inhibiting other microorganisms’ growth [70]. Additionally, *Lactobacilli* strains produce medicinal probiotic metabolites and bacteriocin BACs, often used as biological preservatives in the food industry, arousing people’s attention [114]. These metabolites have antagonistic activity against bacteria genetically similar to producing bacteria, which are immune to their own BACs. BACs have also been considered biologically active molecules with potential activities for human health, such as use as antiviral and anticancer drugs. BACs are extracellular antimicrobial peptides synthesized by ribosomes. They have extensive antibacterial activity and are a safe alternative to antibiotics. As a result, the shelf life of naturally fermented foods has increased. Therefore, screening high-yield BAC probiotic *Lactobacilli* strains from naturally fermented food should be an option. Researchers have also studied the production and characterization of BACs by different probiotics [12]. Additionally, the combined culture of different probiotics may produce new antibacterial products [115,116].

Some reports show that *Lactobacilli* strains have antioxidant activity and can be used as antioxidants in food, stabilizing food’s color, flavor, and taste [80]. Additionally, *Lactobacilli* strains can reduce the oxidative stress injury of Caco-2 cells and improve the antioxidant capacity under oxidative stress. Firstly, the tolerance of *Lactobacilli* strains to hydrogen peroxide was studied [79]. The antioxidant capacity of *Lactobacilli* strains was evaluated by measuring the hydroxyl radical scavenging capacity of cell-free extracts of these strains. These strains can produce metabolites such as superoxide dismutase, glutathione, and extracellular polysaccharide to inhibit oxidation.

### 2.4. Fermentation Performance of Lactobacilli Strains

As a lactic acid starter, *Lactobacilli* strains should be tolerant to harsh conditions, such as temperature changes, osmotic pressure (high fat and protein concentration in milk and meat and high salt in kimchi), and lactic acid accumulation. These characteristics can ensure the rapid adaptation and growth of microorganisms to bring good physical properties and taste to the products. Due to different food components, some of them are used for specific food manufacturing, such as yogurt and cheese (*L*. *delbrueckii*), fermented vegetables (*L*. *plantarum* and *L*. *pentosus*), and fermented meat (*L*. *pentosus*).

The diversity of lactic acid food produced by *Lactobacilli* strains requires that the fermentation characteristics of these strains are different [117]. For example, *Lactobacilli* strains used in meat processing should be able to improve the flavor of end products without producing biogenic amines, because these compounds are produced by the deacidification of free amino acids and have toxic effects on human intestines. Studies have revealed that *Lactobacilli* strains with protein hydrolytic activity [73], which belong to homogeneous fermentation, can significantly reduce the biogenic amines of fermented sausage. The production of bacteriocin by *Lactobacilli* strains, for example, is another feature of evaluating the development of meat products by *Lactobacilli* strains. It can inhibit the growth of pathogenic bacteria and increase the shelf life of products. As mentioned above, the antibacterial activity of *Lactobacilli* strains was screened to resist various pathogens evaluating the production of nisin against *Listeria monocytogenes*, *Clostridium perfringens*, *Bacillus cereus*, and *Staphylococcus aureus* [51,69,70,71].

*Lactobacilli* strains produce large amounts of lactic acid, a non-volatile, odorless compound that contributes to the aroma of the product [118]. Therefore, the production of another fermentation performance flavor molecule was evaluated by gas chromatography–mass spectrometry. The main aroma components include aldehydes, organic acids, higher alcohols, esters, carboxylic acids, and ketones [119,120]. *Lactobacilli* strains convert precursor molecules into aromatic compounds by secreting various extracellular enzymes [73,74,75]. In a protein-rich environment, proteolytic enzymes play a major role in forming aromatic molecules from the amino acids released by complex proteins. For example, milk is rich in casein, and *Lactobacilli* strains used in yogurt and cheese convert these precursor molecules into flavor substances. Lipid degradation also plays a vital role in the aroma formation of fermented meat and dairy products [121].

### 2.5. Health Functions of Lactobacilli Strains

From the functional characteristics, the existence of *Lactobacilli* strains in the intestinal microbiota is related to the host’s health status [122,123]. Clinical research showed that the reduction or increase in the proportion of this genus in the human body would produce health problems, such as irritable bowel syndrome [124], human immunodeficiency virus [125], obesity [126], type 2 diabetes [127], etc. Several strains with probiotic properties of specific metabolites have been successfully included in various functional foods (Figure 3). Although the mechanism of their role is uncertain, the health-promoting effect of these strains is related to the strains themselves and their active metabolic substances (such as extracellular polysaccharide, bacteriocin, polypeptide, short-chain fatty acid, etc.). Several meta-analyses and systematic reviews [128,129,130,131,132,133] support the health effects of probiotic *Lactobacilli* strains and specific metabolites produced by *Lactobacilli* strains in treatment cases, including acute rotavirus diarrhea in children, antibiotic-related diarrhea in children, *Helicobacter*
*pylori* infection, allergic rhinitis, high blood pressure, hyperlipidemia, and other diseases. Additionally, based on its excellent physiological characteristics and probiotic function, *Lactobacillus* has become an important direction in the field of probiotic function. Given the specificity of the *Lactobacilli* strains, the new strains may have potential unique functional characteristics. Table 4 summarizes several vital functions of *Lactobacilli* strains.

### 2.6. Performance Development and Improvement of Lactobacilli Strains

As mentioned above, numerous species of *Lactobacilli* are used in food production (Table 1), including improving traditional food and developing new products. On the one hand, *Lactobacilli* strains can enhance the quality of fermented food and, on the other hand, prolong the storage period of food as a preservative. Therefore, the excellent characteristics of *Lactobacilli* strains are the key to their application in the food industry. However, *Lactobacilli* strains have specificity themselves. Different strains of the same species show significant differences; therefore, new characteristics can be found [138]. Thus, scholars are committed to screening new *Lactobacilli* strains.

On the one hand, the growth of naturally screened *Lactobacilli* strains is limited by physical and chemical factors, such as pH [139,140], oxygen [141,142], osmotic stress [143], temperature [140], carbohydrate substrates[144], and other factors [145,146]. On the other hand, the yields of beneficial metabolites of naturally screened *Lactobacilli* strains, such as lactic acid [147], γ-aminobutyric acid [148,149], extracellular polysaccharide [149], and bacteriocin [19] are relatively low and cannot meet the requirements of industrial production. Therefore, reasonable breeding strategies are used to improve the performance of *Lactobacilli* strains with potential application in the food industry. 

One method is mutagenesis breeding. Mutation breeding of *Lactobacilli* strains can change the genetic structure and function of *Lactobacilli* strains, and then screen mutants to obtain the required high-yield and high-quality strains [150]. It is the most basic modern breeding method. The breeding speed is fast, the cost is low, the time is short, and the method is simple, mainly including physical, chemical, and biological mutagens. Chemical mutagenesis primarily uses nitrosoguanidine, diethyl sulfate, and other chemicals. These chemicals are harmful to the human body. Thus, they are not widely used in the food industry. A limited number of studies focused on the biological mutagenesis of *Lactobacilli* strains, mainly involving transposon mutations [151,152]. Physical mutagenesis of *Lactobacilli* strains commonly uses ultraviolet [153] or microwave radiation [154]. Given the possible tolerance of traditional radiation technology of *Lactobacilli* strains, new mutation technologies, such as heavy ion beam irradiation and plasma mutation breeding, have recently appeared [155,156,157]. The operation of traditional mutation breeding is simple, and the experimental conditions are not high; the mutation is random, and the workload is enormous despite the introduction of high-throughput screening technology in the mutation process [155,158,159].

Another method is metabolic engineering, a continuation, and upgrade of gene engineering technology. This method can directionally change the functional characteristics of *Lactobacilli* strains and compensate for the shortcomings of classical mutagenesis screening [160,161,162,163,164]. The metabolic strategies of *Lactobacilli* mainly focus on the changes in pyruvate metabolism to produce essential fermentation end products, such as sweeteners, spices, aromatic compounds, and complex biosynthetic pathways, leading to the production of extracellular polysaccharides and vitamins [165]. Currently, the most commonly used methods for metabolic engineering of *Lactobacilli* include whole-genome amplification [166], genome shuffling [167,168], and genome editing (plasmid-based homologous recombination, Red/RecET-mediated double-stranded DNA recombination, and single-stranded DNA recombination) [169,170]. However, the safety of these methods for metabolic engineering to change the metabolic characteristics of *Lactobacilli* is worth considering and is not accepted by the European Union [171].

### 2.7. Role of Lactobacilli Strains in Food Production

The primary role of *Lactobacilli* strains in dairy processing (such as yogurt, dahi, kefir, koumiss, and cheese) is not only to improve the nutritional value but also to produce lactic acid, butyric acid, a variety of amino acids, and vitamins and other metabolites, resulting in a unique food flavor. Additionally, these strains use dairy products as a carrier to promote human health due to their probiotic effect [172]. The application of *Lactobacilli* strains to meat products can improve the appearance of meat products, promote the improvement of taste, inhibit the growth of spoilage bacteria, reduce the generation of nitrite and greatly improve the overall quality of meat products [27]. 

In turn, fermented foods as a carrier play a role in transporting and storing these excellent strains. On the one hand, these strains were found in traditional fermented foods, which characterized their excellent properties. On the other hand, these strains were intensively inoculated into conventional fermented food to improve product control. Fermented fruits can be produced by natural fermentation of the surface flora spontaneously formed (such as *Lactobacilli* and *Pediococcus* spp.) or inoculated with fermentation starter (such as *L*. *plantarum*, *L*. *rhamnosus*, and *L*. *acidophilus*). Food nutritionists are developing a new generation of fermented fruit products with special biological and unique sensory characteristics [5,173,174]. Fermented vegetable products can positively impact human health because they are rich in substances beneficial to human beings (such as dietary fiber, minerals, antioxidants, and vitamins). The principle is to use *Lactobacilli* strains attached to vegetables and several artificially selected excellent strains to carry out a series of microbial fermentations and finally obtain the finished pickle. The *Lactobacilli* contained in kimchi can promote human gastrointestinal peristalsis, reduce fat, and enhance immunity [10,120].

The fermentation of probiotic strains with excellent performance has attracted people’s attention. The screened new strains are often used in the development of new products. In recent years, several *Lactobacilli* strains have been widely used in various functional foods due to their unique physiological efficacy and flavor, such as active *Lactobacilli* drinks and solid drinks [119,175]. With the deepening of relevant research, *Lactobacilli* will be used in human health conditioning treatments as a probiotic functional food to a greater extent, and the application direction will be more extensive.

*Lactobacilli* can also be applied to preserving food, such as meat, fruit, vegetables, seafood, etc. These *Lactobacilli* strains are used as biological preservatives due to the following manifestations: (1) produce organic acids, such as lactic acid and acetic acid, to inhibit the growth and reproduction of most spoilage bacteria; (2) H_2_O_2_ production activates the catalase thiocyanate system in milk; (3) produce small proteins or peptides similar to bacteriocin, etc., [176,177].

## 3. High-Density Fermentation

High-density cell culture of *Lactobacilli* strains is a critical step in producing direct vast set starters and a key challenge at the industrial scale. The application of various emerging culture technologies with equipment to culture the bacteria can significantly increase the density of the bacteria compared with traditional culture, thereby increasing the specific productivity of the target product and offering a fermentation process that obtains more bacteria at a lower cost. 

The high-density fermentation methods of the *Lactobacilli* strain mainly involve optimizing medium composition and culture conditions [178]. The current high-density fermentation methods for increasing the concentration of bacteria have certain advantages and disadvantages (Table 5).

**Table 5 foods-11-03063-t005:** The current high-density-culture methods showing advantages and disadvantages.

Current High-Density-Culture Methods	Advantage	Disadvantages
Buffer salt culture [179]	Add a buffer salt that has no effect on the strain or has a growth-promoting impact on the culture medium to improve the buffering capacity of the fermentation broth and control the stability of pH within a specific range, easy to operate.	The buffering capacity of the buffer salt is limited and can only play a role within a specific range.
Chemical neutralization culture [180]	Add lye (such as NaOH, ammonia, and CaCO_3_) to the culture system to control the pH value of the fermentation system, easy to operate.	With the continuous addition of lye and the accumulation of metabolites, too high salt concentration will inhibit the growth of bacteria.
Dialysis culture [181]	Remove part of the small molecular metabolites produced by the bacteria while providing fresh nutrients to the culture solution.	A small processing volume, a long dialysis process, and large equipment investment are also not conducive to industrialization.
Fed-batch culture (non-feedback mode and feedback mode) [182,183,184,185]	Effectively eliminates substrate inhibition and acid inhibition and is simple to operate	Inadequate utilization of nutrients; Limited by container volume
Cross-flow culture [186]	Due to cross-flow filtration, the high viscosity produced by cells is reduced, which is conducive to cell recovery and high concentration	High equipment cost; Requires more professional operators; It is easy to block the membrane module.
Circulating culture (sedimentation, centrifugation, and membrane filtration) [187,188]	Through technologies such as sedimentation, centrifugation, and membrane filtration, the cells are intercepted, the culture medium flows out, and then a certain amount of fresh culture medium is added to obtain high-density cells. It shortens the production time and saves a lot of power, workforce, water, and steam	In the circulation process, the strains quickly degenerate and are polluted, resulting in economic losses; The utilization rate of nutrients was lower than that of batch culture.

Recent advances in high-density fermentation methods showed that the fed-batch culture strategies, including non-feedback modes, such as intermittent feeding, constant feeding, exponential feeding, and feedback mode, are the most widely used for the high-density growth of *Lactobacilli* strains (Table 5). The combination of high-density cell culture methods is the current and future development trend for studying *Lactobacillus*.

## 4. Production of DVS Starter of *Lactobacilli*

Fermented food has experienced the methods of natural fermentation, inoculation fermentation, and pure culture fermentation, that is, a directed vat set (DVS) starter developed today [189]. A DVS starter has a stronger fermentation activity and higher viable bacteria (>1 × 10^11^ CFU/g). This starter can be directly inoculated during the production of lactic acid fermented food, has no intermediate subculture process, is easy to use, and is a new commercial production strain with a stable product quality [189]. The DVS starters of *Lactobacilli* are dry powders made from the selection of excellent strains, proliferation, and culture in a liquid medium, concentration, and separation, combined with a biological protective agent and drying (Figure 4). Fermented foods often require more than one strain. For example, yogurt usually includes *Streptococcus thermophilus* and *L*. *bulgaricus*. Additionally, adding other strains enhances fermentation performance and potential probiotic properties. Therefore, many researchers have studied the compound DVS starters for manufacturing various foods (e.g., yogurt, cheese, fermented meats, and vegetables) and probiotic products. 

There are still deficiencies in production technologies for DVS starters, manifested in four aspects: strain breeding, optimization of culture medium, high-density culture, and cell drying. Drying is the last and most critical step in preparing DVS starters with high activity and stability. The most common methods for cell drying of *Lactobacilli* strains are freeze and spray drying [190]. Freeze-drying technology is the best choice for producing bacterial powder because microorganisms are sensitive to heat. The process utilizes the phase change of water when it is lower than the triple point, freezing the free water in the culture solution and sublimating the ice crystals into water vapor under a vacuum [190]. After the freeze-dried cells reach low temperature, dryness, vacuum, and other conditions, their metabolic activities stop, and they are in a dormant state and can be stored for a long time. During the freeze-drying process, the bacterial cells will be damaged due to freezing and drying. Therefore, the appropriate *Lactobacilli* species must be selected. The study showed that the freeze-dried survival rate of *Lactobacilli* is lower than that of *Lactococcus*, and this can be observed at the end of logarithmic growth and at the same time [190]. Additionally, the culture medium, freezing temperature, and cell membrane composition affect the freeze drying of bacteria [191]. Even if low product temperatures are applied, varying degrees of viability recovery have been reported for freeze-dried bacteria.

Another drying method is spray drying. Spray drying is high-speed centrifugation or high-pressure method, spraying the starter containing the bacteria into extremely fine droplets in a dry heat medium, allowing the water to evaporate quickly, thereby obtaining a dry bacterial sample [192]. Compared with freeze-drying and freezing methods, spray-drying technology has the advantages of simple equipment, low cost, and suitability for large-scale production, transportation, and storage of probiotics. However, this method will cause specific damage to bacteria and affect their survival rate due to their exposure to dry heat and rapid dehydration. Some effective heat protection agents such as lactose, trehalose, whey protein isolate, and skimmed milk [8,193,194] are selected to protect *Lactobacilli* cells during the spray drying process. 

Recently, some emerging drying technologies for the production of DVS starters of *Lactobacilli*, such as low-temperature spray-drying technology [195], freezing granulation drying technology [196], and spray freeze-drying [197], have been developed, which contribute to solving the problem of low survival rate.

Producing probiotics with high activity is the key to ensuring product quality and enhancing market competitiveness. In the traditional batch process, each operation unit is discontinuous, increasing the risk of bacteria exposure. Therefore, the development trend of the probiotic industry involves the development of production equipment for automatic continuous cell culture, separation, drying, and other operating units to realize continuous production. Combined with the advantages of freeze and spray drying, the equipment and technology for developing low-temperature spray drying, freezing granulation drying, and spray freeze-drying have potential for large-scale application [189,190,196,197]. The preparation process of cryogenic bacteria is also one of the essential directions for preparing highly active probiotics [198]. The development of packaging machines and related supporting equipment under low temperatures is a problem that needs further solving in producing cryogenic bacteria in China.

## 5. Conclusions and Outlook

A large number of published studies showed that bacteria belonging to *Lactobacilli* could spontaneously form a microbial group in fermented food, such as in food processing micro-factories, continuously transporting substances beneficial to human health. With the advancement of modern molecular biotechnology, various *Lactobacilli* species in fermented food have been identified, several complete genomes have been obtained, and the probiotic mechanism of *Lactobacilli* species has been revealed. The separation, identification, and characterization of *Lactobacilli* species from various foods showed differences in fermentation performance, acid production, acid tolerance, bile resistance, antimicrobial activity, cholesterol-lowering ability, and antibiotic resistance. Highly active *Lactobacilli* species preparations can be obtained by high-density cultivation and used as DVS starters to realize the industrialization of fermented foods.

Screening out excellent *Lactobacilli* strains from various food resources has always been an important research topic. At present, diverse *Lactobacilli* species are used in the food industry. Foods fermented by *Lactobacilli* species not only improve the original flavor of the food but also increase the food’s nutritional value, which has a good health effect on the human body. In the future, we can use the advantages and characteristics of *Lactobacilli* species fermentation, conduct an in-depth study of the health functions of *Lactobacilli* species and protection technology of high-efficiency live bacteria protection technology, and develop new products with health-care functions. In addition, with the increase in the maturity of multi-omics technology, the application of multi-omics technology in *Lactobacilli* species is increasingly being favored and valued. The comprehensive applications of genomics, transcriptomics, proteomics, and metabonomics can reveal the genetic information of *Lactobacilli* species, analyze physiological and metabolic mechanisms of *Lactobacilli*, penetrate the adaptation mechanism of *Lactobacilli* species to physiological and environmental changes, and explore the molecular mechanism of beneficial functions of *Lactobacilli* species. All of these will promote the rapid development of the *Lactobacillus* fermenting food industry.

## Figures and Tables

**Figure 1 foods-11-03063-f001:**
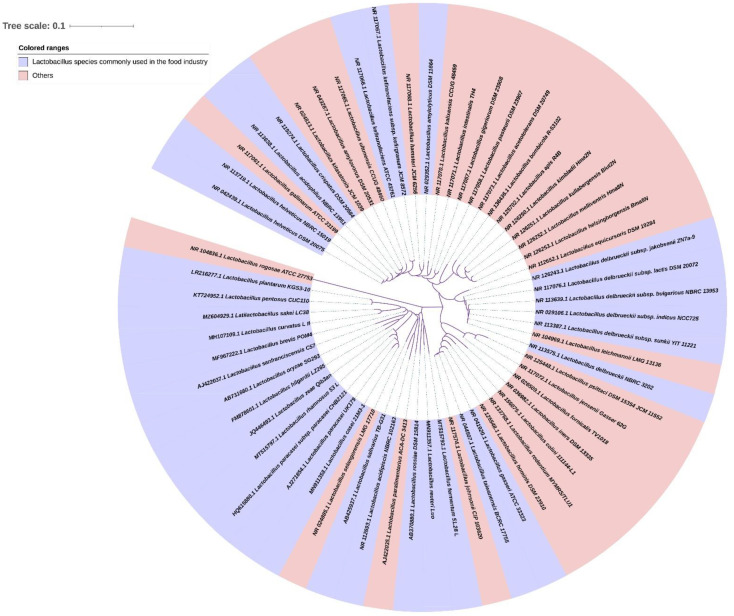
Neighbor-joining phylogenetic tree based on 16S rRNA gene sequences showing the relationship between known species in *Lactobacillus*, produced and rendered by an online tool—Interactive Tree of Life (https://itol.embl.de/, accessed on 21 March 2022) [43]. The lilac background color represents *Lactobacillus* species commonly used in the food industry (including fermented fish products, fermented dairy products, fermented soy products, fermented starch foods, fermented fruit and vegetable, fermented meat products, etc.), and the pink background color represents others (including human gastrointestinal tract, vagina, oral cavity, etc.).

**Figure 2 foods-11-03063-f002:**
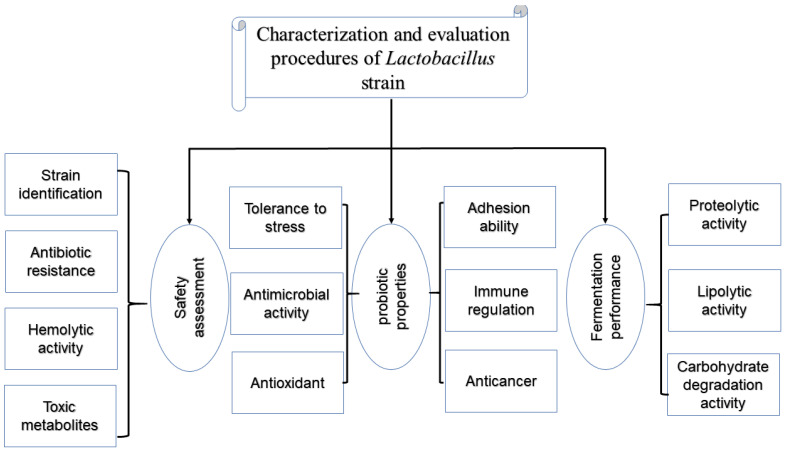
Characterization and evaluation procedures of *Lactobacilli* strains.

**Figure 3 foods-11-03063-f003:**
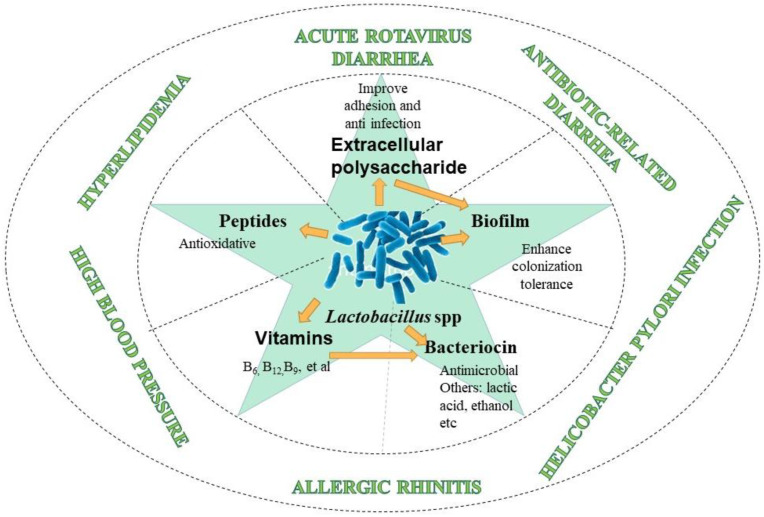
Probiotic properties of *Lactobacillus* spp.

**Figure 4 foods-11-03063-f004:**
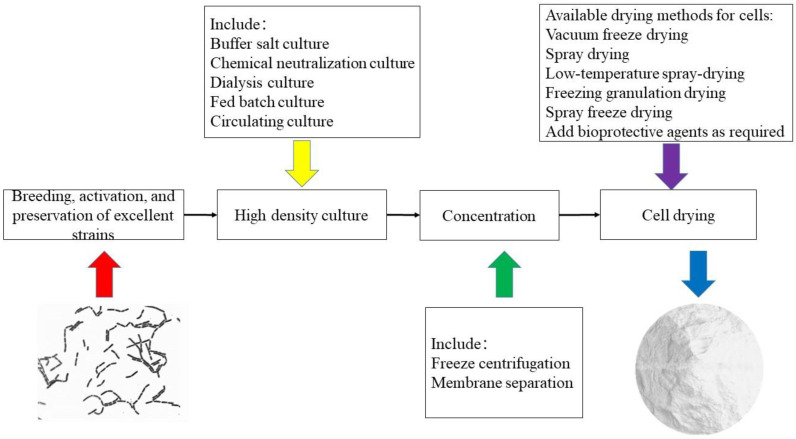
Production process of DVS starters.

**Table 1 foods-11-03063-t001:** *Lactobacilli* species in fermented foods.

Fermented Products	*Lactobacilli* Species Present	References
Fermented fish products	*L. acidipiscis*, *L. brevis*, *L. delbrueckii*, *L. fermentum*, *L. pentosus*, *L. plantarum*, *L. versmoldensis*	[25,26,27]
Fermented dairy products	*L. delbrueckii*, *L. bulgaricus*, *L. fermentum*, *L. plantarum*, *L. kefiri*, *L. paracasei* subsp. *Paracasei*, *L. rhamnosus*, *L. curvatus*	[6,28,29,30,31,32,33,34]
Fermented soy products	*L. amylophilus*, *L. buchneri*, *L. delbrueckii*, *L. fermentum*, *L. paracasei*, *L. plantarum*, *L. salivarius*	[35,36,37]
Fermented starch foods	*L. acidophilus*, *L. brevis*, *L. casei*, *L. bulgaricus*, *L. oryzae*, *L. pentosus*, *L. reuteri*, *L. rhamnosus*, *L. rossiae*, *L. sakei*, *L. curvatus*, *L. panis*, *L. sanfranciscensis*	[38]
Fermented fruit and vegetable	*L. acidophilus*, *L. brevis*, *L. casei*, *L. fermentum*, *L. pentosus*, *L. plantarum*	[28,39,40]
Fermented meat products	*L. sakei*, *L. curvatus*	[41,42]

**Table 2 foods-11-03063-t002:** Characteristics of *Lactobacilli* strains.

Characteristic	Assays	Representative References
Safety	Strain identification (including physiological and biochemical tests, molecular level)	[44,45,46,47]
	Antibiotic resistance	[48]
	Hemolytic activity	[49]
	Determination of potential metabolites (enzyme production, toxin production, production of biogenic amines)	[50,51,52]
Tolerance to stress	Low pH and bile (for example, artificial gastric and pancreatic juices and GIT simulators)	[17,53,54,55,56,57,58]
	Growth environment (for example, nutrition substrate, osmotic pressure, light, temperature, oxygen)	[59,60,61,62]
Adhesion ability	Cell surface hydrophobicity	[63]
	Adhesion to mucus (for example, adhesion to mucin)	[64,65,66]
	Adhesion to Caco-2/TC7 cells	[67,68]
Antimicrobial activity	Production of antimicrobial metabolites such as lactic acid and bacteriocin against pathogenic bacteria (e.g., streak methods, disk diffusion methods, turbidimetric assays, biofluorescence analysis)	[69,70,71]
	Autoaggregation, Coaggregation	[18,72]
Technological properties	Proteolytic activity (e.g., production of various proteases)	[73]
	Lipolytic activity (e.g., production of lipases)	[74]
	Carbohydrate degradation activity (e.g., production of various glycosidases, amylases, cellulases)	[75]
Reduce cardiovascular disease	Cholesterol degradation tests (e.g., Bile salt hydrolase activity)	[76]
	Metabolites such as peptides inhibit the ACE activity	[77,78]
Antioxidant	Tolerance to hydrogen peroxide	[79]
	Metabolites such as the antioxidant activity of extracellular polysaccharides, peptides	[78,80]
Anticancer	Ames test	[81]
	Comet assay	[82]
	Nitrosamine degradation assay	[9]
	Inducing apoptosis of cancer cells test	[83]
Additional characteristics	Conjugated linoleic acid test	[84]
	The removal of heavy metals	[85]
	β-Galactosidase activity analysis	[86]
	Determination of oxalic acid degradation	[87]
	Determination of production of short-chain unsaturated fatty acids and vitamins	[88,89,90]

**Table 3 foods-11-03063-t003:** Molecular approaches used in discrimination among the genus *Lactobacillus*.

Methods Used	Comments	Species Identified and Source	Reference
23S rDNA probe	Probes unequivocally differentiated *L. acidophilus* and *L. plantarum* isolates.	*L. acidophilus, L. pentosus, L. plantarum* species isolates from feed supplements or starter products	[92]
Ribotyping	Good discrimination at strains level based upon differences in rRNA.	Some *L. paracasei* ss. *paracasei* strains as the dominant ones from raw milk cheeses	[93]
RAPD	Good discrimination at strains level.	*L. plantarum* 2035 and *L. plantarum* ACA-DC 2640 isolated from Feta cheese	[94]
Species-specific PCR (plantaricin biosynthesis protein gene)	Rapid and preliminary screening of *L. plantarum* from large vegetable samples before performing a battery of phenotypic and molecular methods.	*L. plantarum* from vegetable samples	[95]
Species-specific PCR using 16S rRNA or unique genes primers	Successful in the species detected in 17 products matched those indicated on their labels, whereas the remaining products contained species other than those appearing on the label.	Some *Lactobacillus* spp., 19 probiotics and 12 dairy products	[96]
Genus- and species-specific PCR, multiplex PCR, real-time HRM analysis, RFLP-PCR, rep-PCR, RAPD-PCR, AFLP-PCR, and proteomic methods such as MALDI-TOF MS typing and SDS-PAGE fingerprinting	Multiplex PCR and MALDI-TOF MS were the most valuable methods to identify the tested bacteria at the species level. At the strain level, the AFLP-PCR method showed the highest discriminatory power.	*L. casei* group, two international collections of microorganisms—the Japan Collection of Microorganisms (JCM) and Belgian Coordinated Collections of Microorganisms (BCCM)	[97]
Comparative sequence analysis, stretches of *rec A* gene	Successful in a clear separation of all type strains in distinct branches; identification of *L. casei* ATCC 393 and *L. casei* ATCC 334 as *L. zeae* and *L. paracasei*, respectively.	*L. casei*, *L. paracasei* (both subspecies), *L. rhamnosus*, *L. zeae*, strains from a commercial probiotic product.	[98]
16S ARDRA, RAPD, Eco RI ribotyping	13 wine strains typed as *L. paracasei/casei*, based on similar band pattern as *L. paracasei* type strain and *L. casei* ATCC 334.	*L. casei*/*L. paracasei* from wine	[99]
PFGE	Good discrimination at strain level based upon different bacterial strains.	The strains of *L. plantarum* isolated from the different fermented foods	[100]
One-step PCR-based, using 16S rRNA genes primers	Successful differentiation among 10 common lactic acid bacteria at the species level.	*L. delbrueckii* and others from fermented milk	[101]
16S ARDRA, ribotyping, RAPD	Only RAPD and ribotyping could discriminate between the type strains of both species.	*L. plantarum*, *L. pentosus*, Wine isolates	[99]
PCR-ARDRA (Taq I), RAPD	ARDRA and RAPD approaches may demonstrate a robust efficiency in the discrimination of unknown isolates.	*L. acidophilus*, *L. planetarum*, and *L. fermentum* from abomasums driven rennet	[102]
Repetitive-element PCR	Could rapidly and easily differentiate *L. brevis* species at strains level.	The closely related strains of *L. brevis* species	[103]
Multi-locus sequence typing (MLST) and multiplex RAPD-PCR	Targeting different genetic variations under the combination of MLST and multiplex-RAPD analysis	*L. sanfranciscensis*, Chinese traditional sourdoughs	[104]
PCR-DGGE, length-heterogeneity PCR (LH-PCR)	Good discrimination at strains level.	Type and reference strains of *L. brevis* DSMZ 20556 and *L. plantarum* DSMZ 2601	[105]
FISH	Rapid and accurate way to identify and quantify bacterial species.	*L. plantarum* (Probiotic products)	[106]

**Table 4 foods-11-03063-t004:** Some crucial functions of *Lactobacilli* strains.

Functional Properties	Example	Reference
Regulating immune system	Jang et al. evaluated immunometabolic functions of *L. fermentum* strains (KBL374 and KBL375) isolated from the feces of healthy Koreans.	[133]
Regulating the balance of blood glucose, blood lipid, and blood pressure	Li et al. found that *L. plantarum* X1 can alleviate the symptoms of diabetes by improving the level of short-chain fatty acids in type 2 diabetic mice.	[134]
Antimicrobial activity	Lim et al. found that *L. paracasei* BK 57 has antagonistic effect on Helicobacter pylori and can be used as potential antibiotics.	[135]
Lower blood pressure	Ong et al. found that *L. paracasei* can isolate and purify ACE inhibitory peptides from cheddar cheese.	[31]
Antitumor	Rajoka et al. found that the antiproliferative activity of the fermentation supernatant of *L. paracasei* SR 4 on cervical cancer cells was up to 89%. *L. paracasei* showed high anti-cancer activity by promoting the up-regulation of BAX, BAD, caspase3, caspase8, and caspase9 genes and down-regulating the expression of the Bcl-2 gene.	[136]
Antioxidant	Suo et al. found that *L. paracasei* ybJ 01 can significantly improve D-galactose, induced the ability of serum superoxide dismutase (SOD), glutathione peroxidase and total antioxidant in mice, and inhibited the production of malondialdehyde.	[137]

## Data Availability

No new data were created or analyzed in this study. Data sharing is not applicable to this article.

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
