# Peer review of "Characterization, High-Density Fermentation, and the Production of a Directed Vat Set Starter of Lactobacilli Used in the Food Industry: A Review"

_foods, 2022, doi:10.3390/foods11193063_

Round 1

Reviewer 1 Report

This work is a review paper on lactic fermentation bacteria and their production, it is known that these microorganisms are well known and widely described. The work does not add new value except as source material, applicable as data collection. It should not be published in significant journals with high impact factor.

Author Response

Response to Reviewer 1 Comments

1.

English language and style

( ) Extensive editing of English language and style required

( ) Moderate English changes required

( ) English language and style are fine/minor spell check required

(x) I don't feel qualified to judge about the English language and style

Response: Thanks.

2.

Is the work a significant contribution to the field?               3

Is the work well organized and comprehensively described?          3

Is the work scientifically sound and not misleading?3      

Are there appropriate and adequate references to related and previous work?        3

Is the English used correct and readable?3

Response: Thanks.

3.

This work is a review paper on lactic fermentation bacteria and their production, it is known that these microorganisms are well known and widely described. The work does not add new value except as source material, applicable as data collection. It should not be published in significant journals with high impact factor.

Response: Thanks. We have highlighted (in Bold) the content of interest to readers in the revised manuscript. The manuscript mainly focuses on reviewing lactobacilli as functional starter cultures in the food industry, including different molecular techniques for identification at species and strains level, methods for evaluating lactobacilli properties, enhancing their performance and improving the cell density of lactobacilli, and the production techniques of DVS starter of lactobacilli strains. Moreover, this review further discussed the existing problems and future development prospects of lactobacilli in the food industry. The viability and stability of lactobacilli in the food industry and gastrointestinal environment are critical challenges at the industrial scale. The new production equipment and technology of DVS starter of lactobacilli strains will have the potential for large-scale application, for example, developing low-temperature spray drying, freezing granulation drying, and spray freeze drying. Therefore, this manuscript provides information of interest to food researchers.

Reviewer 2 Report

In this review, the authors discuss the characterization, the high-density fermentation, and the production of directed vat set starter of lactobacilli used in the food industry. The topic is interesting from a scientific point of view and the discussion is detailed and informative. However, there are a few issues that require the authors’ attention.

1.      Lines 45-54: References are needed per sentence. Especially for the food examples.

2.      Lines 55-76: See comment above. More references are needed.

3.      Lines 63-66: This sentence looks out of place. I believe a new paragraph briefly discussing the most common methods for long term storage of starter cultures could be of use. Alternatively, it could be completely removed since the authors indeed discuss these technologies later on in-text.

4.      Table 1 can be better organized per food category (e.g. fermented drinks, dairy, meat & fish, cereal, etc.). In this way it would be easier for the reader to locate certain food categories and the microorganisms used.

5.      Line 112: Please give a few examples per category.

6.      Table 3 should be reorganized. I believe another column should be added to the right just for references (just like table 2).

7.      Also, in my opinion a Table summarizing the molecular methods used for strain identification in the food industry would be really practical. Each method (pcr reaction, RAPD, 16S RNA sequence determination, bacterial DNA restriction endonuclease analysis, proteomics, metabolomics or other technologies), the microorganism detected, and the food application should be mentioned.

Author Response

Response to Reviewer 2 Comments

1.

English language and style

( ) Extensive editing of English language and style required

( ) Moderate English changes required

(x) English language and style are fine/minor spell check required

( ) I don't feel qualified to judge about the English language and style

Response: Thanks, we have done so in the revised manuscript.

2.

Is the work a significant contribution to the field?               3

Is the work well organized and comprehensively described?          3

Is the work scientifically sound and not misleading?3      

Are there appropriate and adequate references to related and previous work?        2

Is the English used correct and readable?4

Response: Thanks, We have made a reference update in the revised manuscript.

3.

In this review, the authors discuss the characterization, the high-density fermentation, and the production of directed vat set starter of lactobacilli used in the food industry. The topic is interesting from a scientific point of view and the discussion is detailed and informative.

Response: Thanks.

Lines 45-54: References are needed per sentence. Especially for the food examples.

Response: Thanks, we have done so.

5.

Lines 55-76: See comment above. More references are needed.

Response: Thanks, we have done so.

Lines 63-66: This sentence looks out of place. I believe a new paragraph briefly discussing the most common methods for long term storage of starter cultures could be of use. Alternatively, it could be completely removed since the authors indeed discuss these technologies later on in-text.

Response: Thanks, we have done so.

Table 1 can be better organized per food category (e.g. fermented drinks, dairy, meat & fish, cereal, etc.). In this way it would be easier for the reader to locate certain food categories and the microorganisms used.

Response: Thanks, we have done so.

Line 112: Please give a few examples per category.

Response: Thanks, we have done so.

9.

Table 3 should be reorganized. I believe another column should be added to the right just for references (just like table 2).

Response: Thanks, we have done so.

10.

Also, in my opinion a Table summarizing the molecular methods used for strain identification in the food industry would be really practical. Each method (pcr reaction, RAPD, 16S RNA sequence determination, bacterial DNA restriction endonuclease analysis, proteomics, metabolomics or other technologies), the microorganism detected, and the food application should be mentioned.

Response: Thanks, we have done so.

Reviewer 3 Report

The research paper foods-1894118- entitled: ”Characterization, high-density fermentation, and the production of directed vat set starter of lactobacilli used in the food industry: a review" is quite interesting.

This review focuses on compiling information from recent years on lactobacilli strains and discussing the existing problems and future prospects for the development of lactobacilli in the food industry. The review describes the properties of the lactobacilli strains, their use in the food industry and the current methods of improving the cell density of the lactobacilli strain. Techniques for the production of the DSV primer from the genus Lactobacillus were also described.

For many years, lactic acid bacteria have inspired researchers to discover newer and newer properties and functional traits of the strains studied. New methods of obtaining and applying them in the industry are also searched for, and new techniques are often not perfect. Taking into account all these aspects, I believe that the work is up-to-date and interesting.

Collecting knowledge from recent years allows you to follow the research of scientists and assess what else is to be done on a given topic. The development of food production technology and the need to search for new techniques always require gathering current knowledge on a given topic and its analysis.

The manuscript in general is good organized. The article is presented in a general, clear and transparent manner. The conclusions are worthy of the evidence and arguments presented. In this review, the authors will address the main question posed. The review is quite interesting and will contribute to updating the literature on the topic.

Comments:  Italicising species and genus name is required (inconsistent currently and sometime spelt incorrectly). Use italics throughout the text for in vitro and in vivo. Some of the references are a bit old - 30% of references are more than 10 years old. In line 75, please replace, " futher" with "further". In line 107, please replace, "is" with"are". In line 114, remove "the". In line 140, correct to "phylogeny". In line 187, correct "1x106 CFU". In line 231, correct to "polysaccharides". In line 233, delete "lactobacilli" (doubled). In line 406, correct to "DVS".   A graphical abstract could support future the study.

Author Response

Response to Reviewer 3 Comments

1.

English language and style

( ) Extensive editing of English language and style required

( ) Moderate English changes required

(x) English language and style are fine/minor spell check required

( ) I don't feel qualified to judge about the English language and style

Response: Thanks, we have done so.

2.

Is the work a significant contribution to the field?               4

Is the work well organized and comprehensively described?          4

Is the work scientifically sound and not misleading?          4

Are there appropriate and adequate references to related and previous work?4     

Is the English used correct and readable?                4

Response: Thanks, we have done so.

3.

The research paper foods-1894118- entitled: ”Characterization, high-density fermentation, and the production of directed vat set starter of lactobacilli used in the food industry: a review" is quite interesting.

Response: Thanks.

4.

This review focuses on compiling information from recent years on lactobacilli strains and discussing the existing problems and future prospects for the development of lactobacilli in the food industry. The review describes the properties of the lactobacilli strains, their use in the food industry and the current methods of improving the cell density of the lactobacilli strain. Techniques for the production of the DSV primer from the genus Lactobacillus were also described.

Response: Thanks.

5.

For many years, lactic acid bacteria have inspired researchers to discover newer and newer properties and functional traits of the strains studied. New methods of obtaining and applying them in the industry are also searched for, and new techniques are often not perfect. Taking into account all these aspects, I believe that the work is up-to-date and interesting.

Response: Thanks.

6.

Collecting knowledge from recent years allows you to follow the research of scientists and assess what else is to be done on a given topic. The development of food production technology and the need to search for new techniques always require gathering current knowledge on a given topic and its analysis.

Response: Thanks.

7.

The manuscript in general is good organized. The article is presented in a general, clear and transparent manner. The conclusions are worthy of the evidence and arguments presented. In this review, the authors will address the main question posed. The review is quite interesting and will contribute to updating the literature on the topic.

Response: Thanks.

8.

Italicising species and genus name is required (inconsistent currently and sometime spelt incorrectly).

Response: Thanks, we have done so.

9.

Use italics throughout the text for in vitro and in vivo.

Response: Thanks, we have done so.

10.

Some of the references are a bit old - 30% of references are more than 10 years old.

Response: Thanks, we have replaced some of them in the revised manuscript.

11.

In line 75, please replace, " futher" with "further".

Response: Thanks, we have done so.

12.

In line 107, please replace, "is" with"are".

Response: Thanks, we have done so.

13.

In line 114, remove "the".

Response: Thanks, we have done so.

14.

In line 140, correct to "phylogeny".

Response: Thanks, we have done so.

15.

In line 187, correct "1x106 CFU".

Response: Thanks, we have done so.

16.

In line 231, correct to "polysaccharides".

Response: Thanks, we have done so.

17.

In line 233, delete "lactobacilli" (doubled).

Response: Thanks, we have done so.

18.

In line 406, correct to "DVS". 

Response: Thanks, we have done so.

19.

A graphical abstract could support future the study.

Response: Thanks, we have done so.

Round 2

Reviewer 1 Report

Dear authors, now the work is more valuable to many scientists from different countries. If other reviewers evaluate the work positively and the esteemed Editor, the work is improved.

Reviewer 2 Report

The authors have addressed all my comments.